# A fully conjugated *meso*-boron-substituted porphyrinoid combining Lewis acidity with redox-switchable aromaticity

Manuel Buckel[1,3], Jonas Klopf[1,3], Johannes S. Schneider[1], Artur Lik[1], Nicolas A. Riensch[1], Ivo Krummenacher[1], Holger Braunschweig[1], Bernd Engels[2] ✉ & Holger Helten [1] ✉

The structural motif of porphyrin is relevant for many essential biological processes and has emerged as a versatile component of functional materials. Here, we introduce a thiophene-based porphyrinogen having electron-deficient boron atoms in all four *meso*-positions. Its fully π-conjugated backbone exhibits effectively concealed antiaromaticity, with locally confined aromaticity to the thiophene units. The macrocycle readily binds fluoride ions, signaled by changes in its photophysical characteristics. Global aromaticity is switched on via facile consecutive (electro)chemical one-electron reductions to give the radical anion and the dianion, the potassium salts both of which were isolated and characterized by single-crystal X-ray diffraction. The bowl-shaped dianion constitutes a tetrathiophene-based porphyrinoid with a metal cation in its coordination sphere. This 18-π-electron macrocycle shows absorption features typical of porphyrins, while its low-energy Q bands are unusually intense in relation to the Soret bands. In addition, it displays fluorescence emission in the near-infrared (NIR) spectral region.

The structural motif of porphyrin (**I**, Fig. 1) plays the key role in many essential processes of living organisms, such as binding and transport of $O_2$ or conversion of light into usable energy[1–6]. In addition, synthetic porphyrins and related macrocycles (e.g., phthalocyanines[7–9] and heteroatom-substituted porphyrinoids[10,11]) have proven highly valuable as active components of functional materials in various fields of application, including organic photovoltaics[1,2], chemosensors[3], photo- and electrocatalysis[4,7], and photodynamic therapy[5,6,8]. While porphyrins and their direct relatives exhibit a closed loop of 18 (or, more generally, [4n + 2]) conjugated π-electrons[12,13], rendering them aromatic, macrocycles that gain such global aromaticity upon two-electron reduction have recently come into the focus of special research interest[14–21]. Such molecules are either antiaromatic in their neutral state ([4n] conjugated π-electrons) or their antiaromaticity is in some way concealed[22], for example, by avoiding the planar

conformation and/or having locally confined aromaticity in constituent subsystems[14–21,23–28]. This can significantly enhance the stability of the macrocycles and make them suitable for practical use, for instance, as battery electrode materials[14,15,17,19] or for electrocatalysis or photoredox catalysis[22].

Embedding trivalent boron with a valence-electron sextet via its empty $p_\pi$ orbital into an organic π system can offer unique properties and functions[29–37]. Incorporation of boron atoms into the core of porphyrins by coordination has been well studied by now[38–40] and subporphyrins and subphthalocyanines are generally available by templating around a boron center[41,42]; however, in the neutral complexes the boron centers are usually tetracoordinate. Very recently, Thilagar, Hickey, and colleagues presented Ni(II) porphyrins with fused B,N heterocycles in their periphery, wherein the boron atoms are tetracoordinate as well[43]. Tricoordinate boron centers have been

[1] Institute of Inorganic Chemistry and Institute for Sustainable Chemistry & Catalysis with Boron (ICB), Julius-Maximilians-Universität Würzburg, Würzburg, Germany. [2]Institute for Physical and Theoretical Chemistry, Julius-Maximilians-Universität Würzburg, Würzburg, Germany. [3]These authors contributed equally: Manuel Buckel, Jonas Klopf. ✉e-mail: bernd.engels@uni-wuerzburg.de; holger.helten@uni-wuerzburg.de

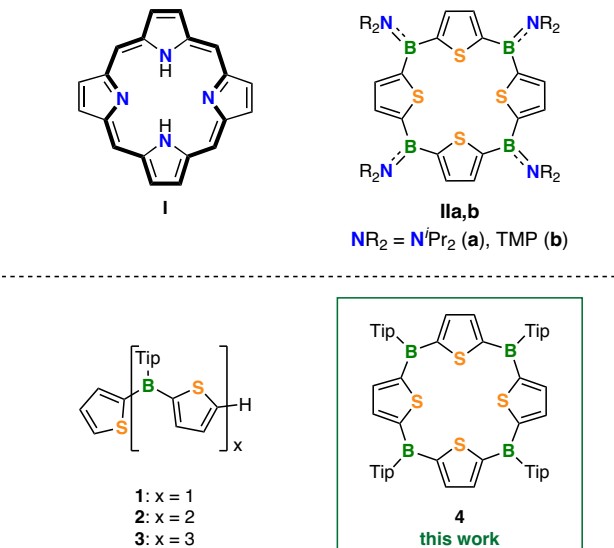

**Fig. 1 | Porphyrins vs. boraporphyrinoids.** Structural motif of porphyrin (**I**, the parent derivative, porphin), with its aromatic conjugation pathway highlighted in bold, and the previously reported boron-bridged tetrathiaporphyrinogens **IIa,b** (TMP = 2,2,6,6-tetramethylpiperidin-1-yl)[62,64]. Oligo(thienylborane) **1**–**3** and the macrocycle **4** reported herein are stabilized by bulky Tip groups (Tip = 2,4,6-tri-*iso*-propylphenyl), which maintains the electron-deficiency of the boron centers.

appended to porphyrins in lateral positions[44–46]. Porphyrin derivatives fused with rings that comprise tricoordinate boron were recently reported by Osuka, Kim, and Yorimitsu[47,48] as well as Pawlicki and co-workers[49]. Omori and Shinokubo incorporated a trivalent boron atom into a *meso*-position of a Ni(II)-coordinated corrole, which gave rise to antiaromatic character[50]. Noteworthy in this context is also a larger conjugated cyclophane bridged via Lewis acidic boron atoms presented by Chen and Jäkle[51], as well as a series of macrocycles that feature boron or boron along with nitrogen atoms in bridging positions, subsequently published by their and other groups[51–59]. We recently added a macrocycle with four direct B=N linkages[60], and Wagner and Anstöter reported annulenes comprising a closed loop of B–E (E=N, O) bonds[61]. Incorporation of boron atoms into the porphyrin framework, while retaining their electron-deficient character, however, poses some difficulties. Porphyrinoids with a boron-doped backbone that comprises an aromatic system of 18 macrocyclic conjugated π electrons remain an elusive endeavor.

In 1998, Corriu, Douglas, Siebert et al. presented the boron-bridged tetrathiophene macrocycle **IIa**[62,63]. Siebert and co-workers subsequently added another derivative, **IIb**[64]. The authors termed these species boron-bridged tetrathiaporphyrinogens. This implies that they should be precursors to (heteroatom-modified) porphyrins. Consequently, upon transformation into the latter, they should gain a globally aromatic electronic structure as it is characteristic for the porphyrin system. The porphyrinogens **IIa,b** comprise a closed cycle of sp² hybridized C and B atoms. If each thiophene unit would contribute 4 electrons to a global π-system in **IIa,b**, this would sum up to 16 cyclic-conjugated π-electrons, rendering these species antiaromatic. However, **IIa,b** showed no evidence of macrocyclic π-conjugation. They were found to be colorless, and in their solid-state structures the constituent thiophene rings were heavily twisted (e.g., in **IIb**[64], with respect to the mean-plane formed by the four *meso*-boron atoms, by 62–66°). Indeed, their structures resemble that of the tetrathiaporphyrinogen by Vogel et al., in which bridging CH₂ groups prevent π interaction between the thiophene rings. In that case, aromatization to a tetrathiaporphyrin dication was achieved by oxidative dehydrogenation[65]. It is conceivable that tetraboraporphyrinogens **IIa,b** may be transformed into an 18 π-electron aromatic porphyrinoid

system via two-electron reduction – although this has not been demonstrated yet. The observed backbone twisting in neutral **IIa,b** might result from their tendency to conceal their antiaromatic character that they would otherwise have in their planar form. Another possible explanation, however, is that the twisting is due to the π-backbonding from the exocyclic amino nitrogen atoms (as indicated by the dashed bonds in the representation for **IIa,b** we chose here). This might largely suppress any possible π-electron delocalization within the macrocycle across the boron atoms. It has also been previously noted that the stabilizing amino groups limit, if not compromise, the Lewis acidity of the boron centers in **IIa,b**[56]. Attaching *ortho*-substituted aryl groups is another established strategy to stabilize tricoordinate boron[66]. The bulky substituents provide kinetic stabilization, while at the same time maintaining the electron-deficient character of the boron center. For example, we applied 2,4,6-tri-*iso*-propylphenyl (Tip) groups to produce very robust thienylborane oligomers **1**–**3** as well as corresponding poly(-thienylborane)s[67,68].

These considerations motivated us to target the boron-bridged porphyrinogen **4** with Tip groups on boron. Herein, we present the synthesis and comprehensive characterization thereof, as well as its follow-up redox chemistry and anion binding. Our studies reveal that the neutral macrocycle is globally non-aromatic, while aromatic character therein is confined to the individual thiophene rings. Nevertheless, it shows extended π-electron delocalization over the entire macrocyclic backbone, as evidenced by spectroscopic and structural features, and supported by quantum chemical calculations. **4** proved to be electron-deficient and behaved as Lewis acid towards fluoride ions. We demonstrate successive one- and twofold reductions to the radical anion **4**·⁻ and further to the dianion **4**²⁻, the potassium salts of which we were able to isolate and fully characterize, including single-crystal X-ray diffraction. Both anions are globally aromatic. While neutral **4** constitutes a fully π-conjugated backbone-boron-doped porphyrinogen, the dianion **4**²⁻ is a true porphyrinoid featuring boron atoms in its framework. It shows weak NIR emission ($\Phi_f = 0.01$) and the typical porphyrin-like absorption behavior, though with intense Q bands.

## Results and discussion
### Synthesis of porphyrinogen 4 and chemical and electrochemical reduction studies to give charged porphyrinoids 4·⁻ and 4²⁻
We accomplished the synthesis of the neutral tetraboratetrathiaporphyrinogen **4** via two different approaches (Fig. 2). The first one involved a [2 + 2] macrocyclization reaction of dilithiated compound **1** with **7** in a dilute solution in THF (0.01 mol L⁻¹). To remove remaining boron-bound methoxide from the macrocyclic product, we subsequently added TMS-Cl to the reaction mixture. This afforded **4** after work-up as a yellow solid in 27% isolated yield. The second approach involved a [3 + 1] macrocyclization in the final step with dilithiated compound **2** and one equiv. of **10** at high dilution conditions (0.01 mol L⁻¹). After analogous work-up we obtained **4** in decent yield of 61% – overall in a shorter time and substantially higher yield than via the first route. The constitution of **4** was unambiguously ascertained by the combination of multinuclear NMR spectroscopy and high-resolution mass spectrometry, which also ruled out the additional presence of macrocycles of larger or smaller sizes (SI, Fig. S21); elemental analysis yielded satisfactory results. The ¹¹B{¹H} resonance at 57 ppm is in a similar range with the corresponding resonances of the shorter triarylboranes **1** (56 ppm), **2** (58 ppm), and **3** (57 ppm)[68]. The ¹H NMR signal of the *β*-thiophene proton is with 7.63 ppm at somewhat lower field than that of unsubstituted thiophene ($\delta = 7.18$ ppm)[69] and in same range with the corresponding protons in the thienylboranes **1**, **2**, and **3** ($\delta = 7.98 - 7.31$ ppm)[68]. Single crystals suitable for X-ray diffraction were obtained for **2**, **9**, **10** (SI, Figs. S54–S56), and **4** (vide infra).

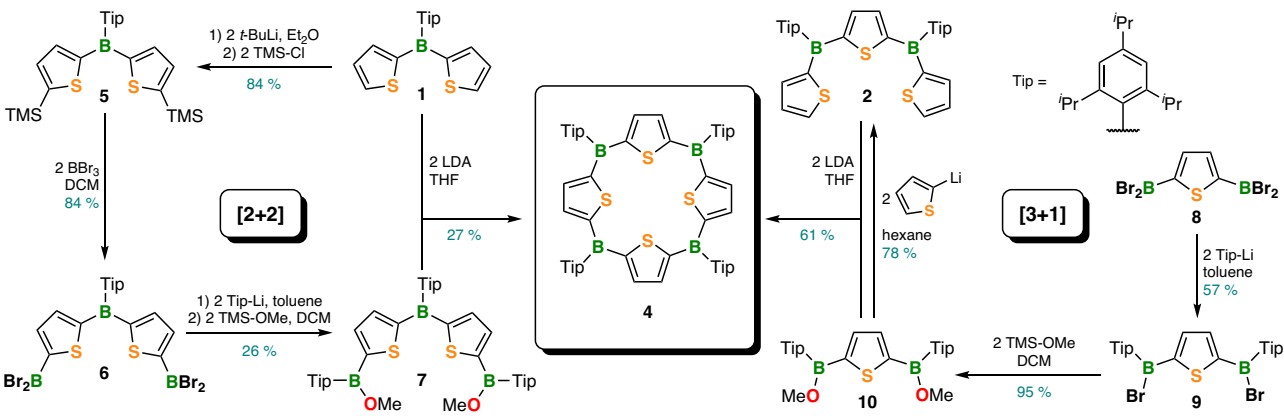

**Fig. 2 | Synthesis of 5,10,15,20-tetrakis(2,4,6-tri-*iso*-propylphenyl)−5,10,15,20-tetrabora-21,22,23,24-tetrathiaporphyrinogen 4.** The macrocycle **4** was obtained via two different routes: [2 + 2] and [3 + 1] hereby denote the number of thiophene units in the reactants **1** and **7** or **2** and **10**, respectively, involved in the ring-closure step.

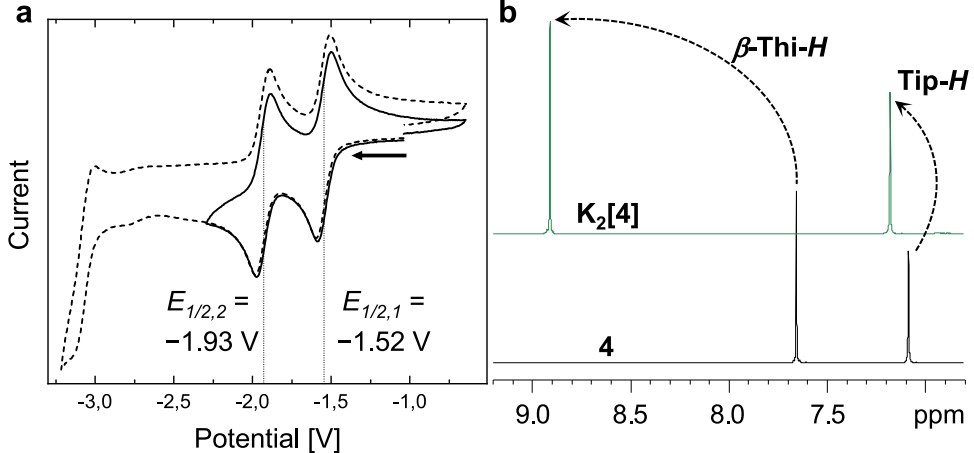

**Fig. 3 | Monitoring of the reduction of 4. a** Cyclic voltammogram of **4** (in THF; supporting electrolyte: [NBu$_4$][PF$_6$]; potential vs. Fc/Fc$^+$; scan-rate: 250 mVs$^{-1}$). Solid line: the first two fully reversible events. Dashed line: full reductive range within the solvent limit. Vertical lines mark the half-wave potentials of the respective redox events. **b** Monitoring of the reduction by observation of the aromatic region of the $^1$H NMR spectra (THF-d$_8$, 300 MHz) (β-Thi-*H*: β-thienyl hydrogen-atoms; Tip-*H*: aromatic hydrogen-atoms of the Tip-substituent).

Cyclic and differential pulse voltammetry (Fig. 3a and SI, Figs. S44–S49) of **4** provided insight into the electrochemical properties of the macrocycle. We detected two distinct reversible reduction events with half-wave potentials of $E_{1/2,1}$ = −1.52 V and $E_{1/2,2}$ = −1.93 V relative to the Fc/Fc$^+$ couple, and an irreversible reduction event starting at about −2.90 V. The first two reductions are substantially anodically shifted compared to that of the shorter open-chain thienylboranes **1** ($E_{1/2}$ = −2.50 V[68]), **2** ($E_{1/2,1}$ = −1.97, $E_{1/2,2}$ = −2.63), and **3** ($E_{1/2,1}$ = −1.80, $E_{1/2,2}$ = −2.30) (SI, Figs. S40–S43).

The finding that the first two redox processes occur at moderate negative potentials vs. Fc/Fc$^+$ encouraged us to explore the reduction also chemically (Fig. 4). We used potassium graphite (KC$_8$) as the reducing agent. The reaction could be well followed by $^1$H NMR spectroscopy (Fig. 3b). Upon addition of one equivalent of KC$_8$, we obtained an NMR silent spectrum, suggesting the formation of a paramagnetic species, presumably the radical monoanion **4**$^{\cdot-}$. Subsequent addition of an excess amount of another equivalent of the reducing agent resulted in the emergence of new $^1$H NMR signals assigned to β-protons of thiophene moieties and the protons of Tip groups, respectively (in 1:1 ratio, as in neutral **4**). The two new signals in the aromatic region are downfield shifted compared to those of **4**. This effect is particularly pronounced for the β-thiophene protons, which shift from 7.63 to 8.91 ppm. Furthermore, the broad signal for the boron atoms in the $^{11}$B{$^1$H} NMR spectrum shifts upfield from 57 to 38

ppm, indicating an increased shielding of the boron nuclei through the added electrons upon reduction (SI, Figs. S11–S14). The newly formed species was identified as the dianionic macrocycle **4**$^{2-}$. We accomplished to isolate its dipotassium salt **K$_2$[4]** as an air-sensitive green crystalline solid in 70% yield by crystallization from a mixture of THF/*n*-pentane at −30 °C. We also achieved to isolate the monoanionic species **4**$^{\cdot-}$ as its potassium salt **K[4]** from a selective comproportionation reaction via mixing of equivalent amounts of **4** and **K$_2$[4]** in THF. Compound **K[4]** crystallized from THF at −30 °C as a red solid in a yield of 71%. Interestingly, the reverse reaction occurs in solvents of low polarity such as *n*-pentane, *n*-hexane, toluene, or benzene. In those solvents the radical anion **4**$^{\cdot-}$ disproportionates back to a 1:1 mixture of **4** and **4**$^{2-}$. This process can be monitored via $^1$H NMR spectroscopy and can be repeated several times by exchanging C$_6$D$_6$ with THF-d$_8$ and vice versa (SI, Fig. S16).

## Structural properties and assessment of the aromatic character of the macrocycles

We were able to grow single crystals suitable for X-ray diffraction analysis for all three macrocyclic compounds **4**, **K[4]**, and **K$_2$[4]** (Fig. 4). The boron atoms in each of them are trigonal-planar coordinated with angle sums close to 360°. The bulky Tip substituents are oriented approximately perpendicularly to the BR$_3$ planes (dihedral angles >74°), thereby providing decent steric protection for the boron

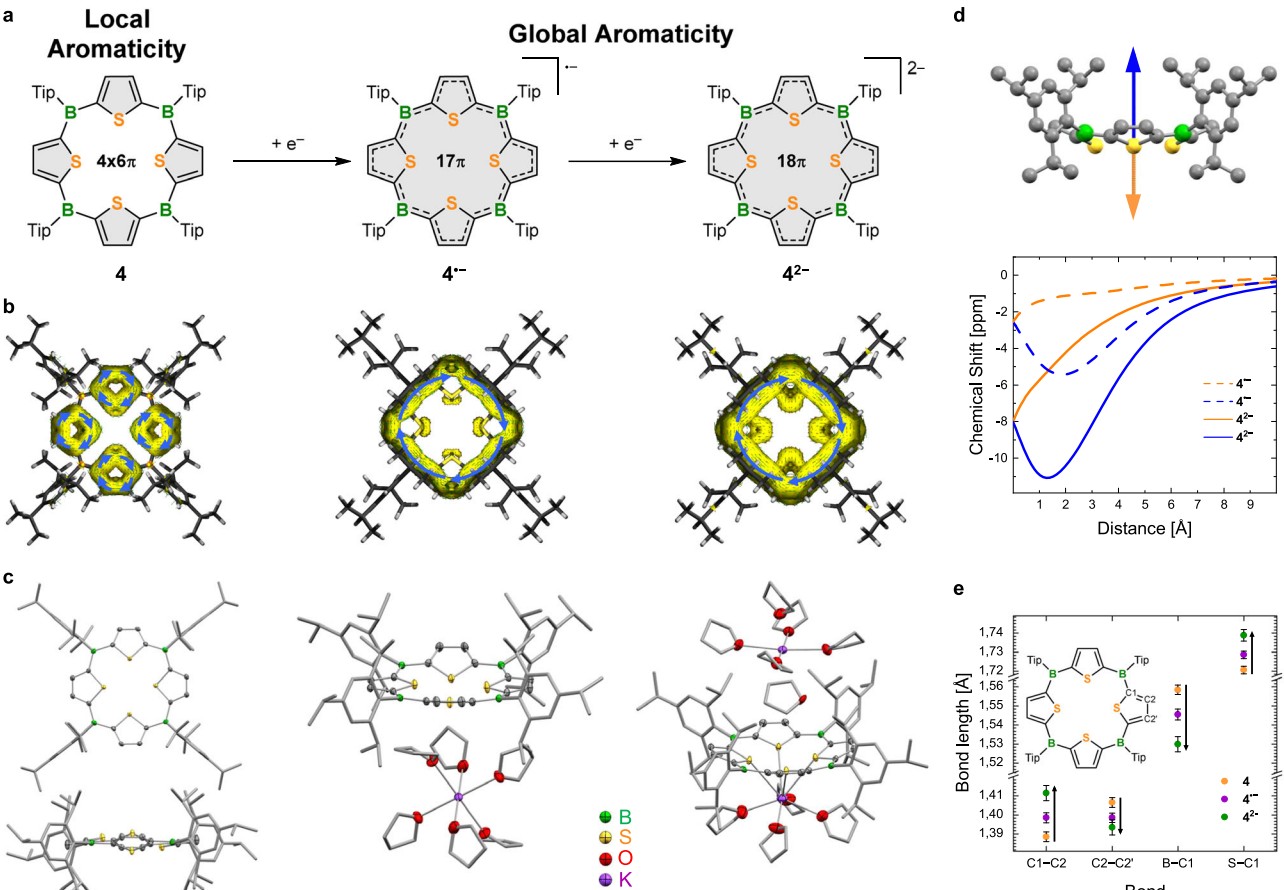

**Fig. 4 | Structural and electronic assessment of the macrocycles. a** Reduction-induced transition from local aromaticity at the thiophene units in **4** to global aromaticity in **4**•⁻ and **4**²⁻. Highlighted in gray are the areas of connected aromaticity. **b** ACID plots of **4**, **4**•⁻, and **4**²⁻ (iso-values: 0.03; blue arrows indicate the direction of the induced ring current). Calculations using ωB97X-D/6-31 + G(d,p). **c** Crystal structures, from left to right: **4** (top and side view), **K[4]**, **K₂[4]**. All atomic displacement ellipsoids are drawn at the 50% probability level. H-atoms are omitted for clarity. Ellipsoids of the Tip substituents and the carbon atoms of all THF molecules are omitted and depicted as capped sticks for clarity instead. **d** NICS scan plots for **4**•⁻ and **4**²⁻ and the associated calculated geometry of the anionic species with indicated scan directions. The geometry of **4**²⁻ is representative for both anionic structures, as their calculated minimum conformation is very similar. Depicted are the isotropic values only (see SI for details, Figs. S68–S71). **e** Graphical illustration of the bond-length trends. For this comparison, the average bond lengths were used, as **4** and **4**²⁻ comprise symmetry-generated parts. Arrows indicate trends from **4** (orange) and **4**•⁻ (purple) to **4**²⁻ (green). The markers at the nodes represent the average margin of error derived from the crystal structures for the respective bond lengths.

centers. The macrocyclic backbones of **4**, **4**•⁻, and **4**²⁻, composed of the four thiophene rings and the boron atoms, are not fully planar, though significantly closer to planarity than the B(NR₂)-bridged tetrathiaporphyrinogens **IIa,b** by Siebert and colleagues (cf. Figure 1)[62,64]. The mean-plane deviations (MPD) of their 24 core atoms are in a similar range (**4**: 0.204 Å, **4**•⁻: 0.180 Å, **4**²⁻: 0.202 Å). While in **4** the thiophenes are twisted out of plane in an up-up-down-down sequence, the thiophene rings of **4**•⁻ and **4**²⁻ all deviate out of the mean-plane in the same direction to form a bowl-shaped structure. In **4** the thiophenes form interplanar angles of 11.3° and 22.3° (two are each generated by symmetry) with respect to the macrocycle's mean-plane. It is noteworthy that the degree of this interplanar twisting in **4** is not only significantly less pronounced than in **IIa,b** (e.g., for **IIb**: 62.1, 63.8, 65.7 and 66.4°)[64] but also closer to that of the aromatic CH-bridged tetrathiaporphyrin dication reported by Vogel et al. (3.7° and 22.8°)[65]. These structural features of **4**, consequently, do not point to a global antiaromaticity in this macrocycle.

In **4**•⁻ one pair of opposing thiophene rings deviates from the macrocyclic mean plane more than the other one, which is quite coplanar with the mean-plane (1.5°/5.5° vs. 20.4°/26.2°). **K₂[4]** crystallized in a tetragonal crystal system wherein all thiophene rings are equally twisted out of plane with an angle of 17.4°. The average dihedral angles between adjacent thiophene units in **4** (27.5°), **4**•⁻ (23.6°), and **4**²⁻ (24.4°) are in the same range with or even slightly smaller than the average angle in the shorter oligothienylboranes **1** (30.4°[68]) and **2** (18.9° and 44.6°; mean values due to disordered terminal thienyl-groups), indicating well-overlapping π-orbitals to aid conjugation throughout the macrocycles. The adjacent sulfur atoms of the thiophenes are distanced about 3 Å from each other in all three macrocycles, while the opposing sulfur atoms are 4.080(1) and 4.511(1) Å apart in **4**, 4.0365(9) and 4.349(1) Å in **4**•⁻, and 4.191(1) Å in **4**²⁻. Although the distance between the opposing sulfur atoms exceeds twice the van-der-Waals radius of sulfur (3.6 Å[70]), there is only a relatively small void in the center of the macrocycles (SI, Fig. S57). The distances of the opposing heteroatoms though are comparable to the nitrogen–nitrogen distances in typical metal-coordinating tetraphenylporphyrins[71] (TTP), for example, Cu-TTP: 3.958 Å, Fe-TTP: 4.039 Å, Pd-TTP: 4.019 Å, and Zn(H₂O)₂-TTP: 4.084 Å. In the solid-state structure of **K[4]**, the potassium cation is located in the periphery of the anionic macrocycle surrounded by six co-crystallized THF molecules. This also applies for one of the cations in the dianionic species **K₂[4]**. The other one is coordinated below the bowl-shaped macrocyclic scaffold by the sulfur atoms of the thiophenes and four additional THF molecules (for more structural details, see SI, Table S4,

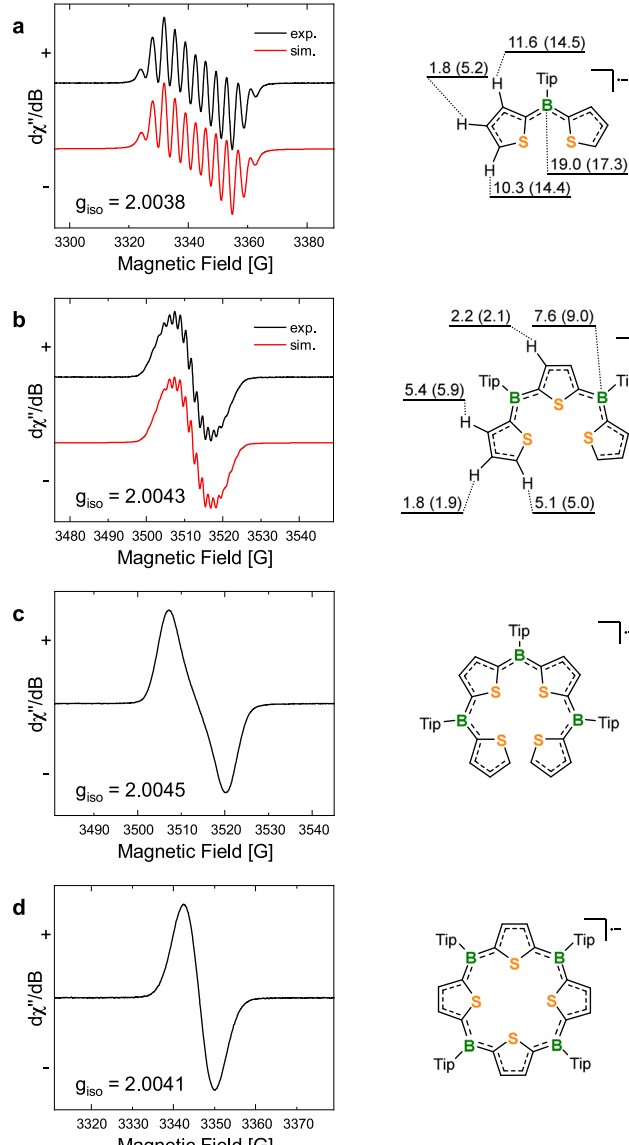

**Fig. 5 | EPR studies. a–d** Experimental EPR spectra of **1**[•–] (a), **2**[•–] (b), **3**[•–] (c), and **4**[•–] (d) in THF (black) and simulated ones (red) for **1**[•–] and **2**[•–] alongside their HFCCs (values in parentheses are calculated at the UωB97X-D/EPR-II level of theory; 6-311 + G(d,p) for the S-atoms). The $a(^1H)$-values in **1**[•–], **2**[•–] were assigned based on the calculated coupling constants as they reflect the same trend, although a clear assignment is not possible due to small differences in certain pairs. Microwave frequencies: 9.37 GHz for **1**[•–], **4**[•–] and 9.85 GHz for **2**[•–], **3**[•–]. All measurements were conducted at room temperature.

Figs. S57−S62). The sulfur−potassium distance of 3.390 Å and the angle between the K−S vector and the vector from sulfur to the C2−C2′ bisector (SI, Fig. S64) of 145.64° suggest sulfur−potassium bonding as this angle is typical for thiophene sulfur-bound metal complexes[72]. Complementary, the inner-thiophene S−C1 bond elongates after reduction from around 1.720 Å in **4** to an average of 1.739 Å in **4**[2−], indicating partial donation of electron density of the sulfur atoms away from the thiophene moiety towards the cation.

There is a clear trend in the bond-length situation from the neutral macrocycle **4** to the reduced species **4**[•–] and further to **4**[2−] (Fig. 4e) that is consistent with a transition from local to global aromaticity upon reduction. It resembles the typical behavior for the transition of porphyrins out of porphyrinogens. The bond lengths between the α-thiophene and the β-thiophene carbon atoms (C1−C2) lengthen in the

process, while the bonds between two β-thiophene carbon atoms (C2−C2′) shorten to form a bond-length equilibration in **4**[•–] up to the point where they become shorter than the C1−C2 bond in **4**[2−] (for numbering of the atoms see Fig. 4e). This apparent increase in the quinoidal character of the thiophene rings points to enhanced π-electron delocalization over the anionic macrocycles. The bond lengths between boron and the α-thiophene carbon atom (C1) also contract after the reduction, from an average of 1.558 Å in **4** to 1.530 Å in **4**[2−].

To gain deeper insight into the electronic structure of these porphyrinoids, we performed DFT studies. The computationally optimized geometries in the gas-phase agree well with the measured crystal structures. To characterize the potential aromatic character of the macrocycles, we conducted anisotropy of the current-induced density (ACID) and nucleus-independent chemical shift (NICS) calculations (Fig. 4 and S1, Figs. S66−S71). The ACID plot for **4** revealed isolated diatropic ring currents within the individual thiophene rings, thus indicating local aromaticity, but no global current over the boron centers. For the reduced species **4**[•–] and **4**[2−], on the other hand, the ACID plots display a clear diatropic ring current around the outer perimeter of the macrocycles. The strength of this effect increases clearly from the mono- (**4**[•–]) to the dianion (**4**[2−]). This global ring current is most likely the origin of the observed downfield shift of the β-thiophene protons in the NMR spectrum of the latter (cf. Figure 3b). NICS scans provide further, more detailed information. For neutral **4**, the NICS scan through the center of the macrocycle perpendicular to its mean-plane does not indicate either significant aromaticity or antiaromaticity. Scans through individual thiophene rings, however, show the typical shape of aromatic systems, consistent with the conclusion drawn from the ACID calculations. For **4**[•–] and **4**[2−], on the other hand, the scans along their z-axes show negative isotropic values throughout, supporting the global aromatic character of the anionic macrocycles. Since they have bowl-shaped conformations, the course of the NICS scan curves differ depending on which side of the curved structure they are observed (SI, Figs. S68, S70). For both **4**[•–] and **4**[2−], the NICS values are more negative inside the bowl (blue)−where the scans exhibit a clear minimum – than outside the bowl (orange; Fig. 4d). The calculated values are more negative for **4**[2−] (−11.1 ppm at the minimum at 1.3 Å) than for **4**[•–] (−5.4 ppm at the minimum at 1.9 Å), consistent with an enhanced global aromaticity in the 18 π electron species **4**[2−].

## EPR spectroscopic analysis
To gain more insight into the nature of the radical anion **4**[•–], we investigated it by EPR spectroscopy in THF. We additionally studied acyclic oligo(thienylborane) radical anions: the monoborane anion **1**[•–], the bisborane anion **2**[•–], and the trisborane anion **3**[•–], for comparison; thus, forming a series with the macrocyclic tetraborane radical anion **4**[•–] (Fig. 5). The EPR resonances of these species range between $g_{iso} = 2.0038$ and 2.0045. The hyperfine couplings with the boron and hydrogen nuclei are well-resolved in the smaller species **1**[•–] and **2**[•–] but diminish with increasing size of the system. From **1**[•–] to **2**[•–] the hyperfine coupling constants (HFCCs) to boron, $a(^{10/11}B)$, decrease from 19.0 to 7.6. The $a(^1H)$ values follow the same trend. Theory reproduced the data quite well, thus allowing us to assign the obtained HFCCs to the specific atoms (Fig. 5a, b). For **3**[•–] and **4**[•–], the hyperfine couplings were unresolved. These observations are consistent with our computationally derived spin densities. With increasing number of borane moieties, the spin density becomes progressively more localized at the boron centers, until it becomes uniformly distributed over all four boron atoms in the macrocyclic aromatic radical anion **4**[•–] (SI, Fig. S87), consistent with a loss of hyperfine couplings.

## Photophysical properties
We investigated the photophysical properties of the macrocyclic compounds **4**, **K[4]**, and **K₂[4]** by UV-vis absorption and fluorescence

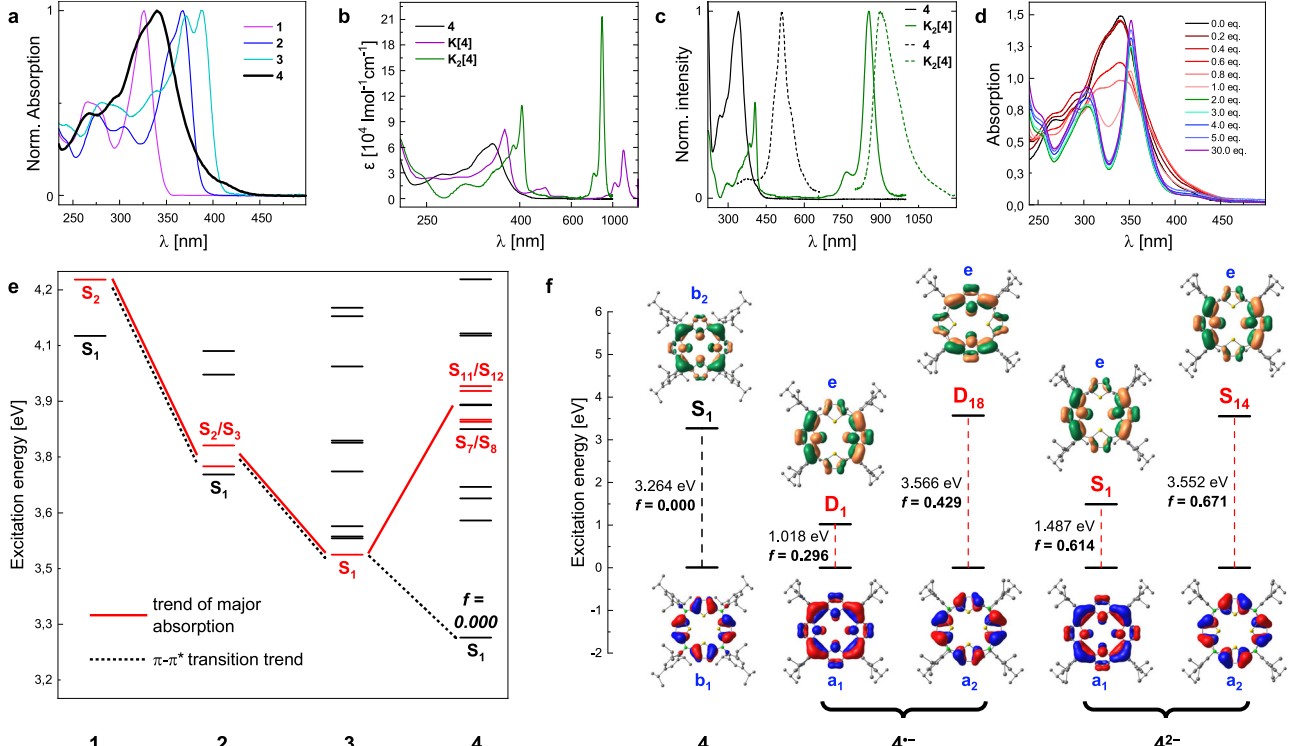

**Fig. 6 | Photophysical data and associated computational studies. a** UV-vis absorption spectra of macrocycle **4** and oligo(thienylborane) **1**, **2**, and **3** in THF. **b** Absorption spectra of **4**, **K[4]**, and **K₂[4]** in THF, plotted with respect to their extinction coefficients against the wavelength in logarithmic scale. **c** Normalized absorption (solid lines) and emission spectra (dashed lines) of **4** (black) and **K₂[4]** (green) in THF. **d** UV-vis absorption spectra of **4** (c = 2.94 × 10⁻⁵ M) upon addition of specified numbers of equivalents of TBAF. **e** Excitation energies for **1–4**. Red colored transitions connected by red lines represent transitions with notable oscillator strengths corresponding to the major absorptions. The dotted black lines indicate the trend of π–π* transitions from **1** to **4**. **f** Natural transition orbitals (NTOs) of **4**, **4·⁻**, and **4²⁻** involved in the S₀ → S₁ transitions and higher-energy excitations responsible for the Soret bands in the absorption spectra of **4·⁻** and **4²⁻**. D₁/D₂ and D₁₈/D₁₉ of **4·⁻** are degenerate (only one of each is shown). The same applies for **4²⁻** for the S₁/S₂ and S₁₄/S₁₅ states. D₁₈ of **4·⁻** involves several NTOs. Due to clarity only the NTOs with the highest percentage contribution are shown (All Data: SI, Computational Results). The symmetries of the depicted NTOs are derived from the optimized structures of the respective compounds and their determined point groups.

emission spectroscopy (Fig. 6). The UV-vis spectra of the acyclic oligo(thienylborane) **1–3**[67,68] are depicted in Fig. 6a in addition to that of **4** for comparison. The former show each a prominent low-energy absorption band, assigned to a π–π* transition within the extended π-system that involves the thiophene rings and all boron centers[67,68]. This band shows successive redshifts with increasing chain length within this series (**1**[67]: λ$_{abs,max}$ = 325 nm < **2**[68]: λ$_{abs,max}$ = 370 nm < **3**[68]: λ$_{abs,max}$ = 387 nm). As the porphyrinogen **4** exhibits an even slightly longer, though cyclic oligo(thienylborane) backbone, it might have been expected to continue the trend of its acyclic relatives. However, the spectrum of **4** shows a comparatively broad main absorption band peaking at 340 nm (ε = 6.4 • 10⁴ L mol⁻¹ cm⁻¹), thus in between the maxima of **1** and **2**. The spectrum additionally shows a tail towards lower energies extending into the visible region. Close inspection of the spectrum let us identify a small shoulder at around 415 nm. This provides an explanation for the bright yellow color of the compound. The macrocycle **4** also shows weak fluorescence emission with a maximum at 511 nm (Φ$_f$ = 0.01; Fig. 6c). The spacing between the absorption and emission maxima is thus comparatively large (561 meV).

Our TD-DFT calculations, indeed, predict the S₁ state of **4** at low energy that fits in the trend of decreasing excitation energies of the oligomers **1–3** (Fig. 6e and S1, Fig. S65, Table S8). Just as in their cases, this process is characterized as a π–π* transition in the extended π-system of its thienylborane backbone. However, in **4** this transition has a vanishing oscillator strength (f = 0.000, HOMO−1/HOMO−8 → LUMO; λ = 379.9 nm). Similar observations have been previously reported for

other bora-macrocycles[51,52,57]. For **4**, we can explain this phenomenon by the high symmetry of its optimized molecular structure (D₂d), since the S₀ → S₁ excitation in this point group is symmetry forbidden because the transition occurs from a natural transition orbital (NTO; Fig. 6f and SI, Fig. S79) of b₁ symmetry to one of b₂ symmetry. Hence, the probability of the lowest-energy π–π* transition becomes zero, i.e., it is dark in the experimental spectrum. The absorption spectrum of **4** is rather dominated by transitions at higher energies. We assign the main band of the experimentally obtained spectrum to excitations into a combination of degenerated S₇/S₈ and S₁₁/S₁₂ states (SI, Table S8 and Fig. S78). Those states have the character of partial intramolecular charge transfer from the Tip substituents to the macrocyclic core. Overall, these findings further corroborate that the porphyrinogen **4** is not antiaromatic. The bathochromic shift of its dark π–π* transition, with respect to that of **3**, moreover reveals that the thiophene rings in **4** are effectively π-conjugated over the boron centers.

We investigated the capability of **4** to act as a Lewis acid through binding of fluoride anions via titration with tetra-*n*-butylammonium fluoride (TBAF) and monitoring of the process by UV-vis and fluorescence spectroscopy. The main absorption band vanishes by successive addition of fluoride, while two new bands emerge, one of which is more red-shifted, at λ$_{max}$ = 352 nm, and the other one appears at higher energy, at λ$_{max}$ = 304 nm (Fig. 6d). Concomitantly, the fluorescence is fully quenched. By means of HRMS, consecutive binding of up to two fluoride anions was detected (SI, Figs. S22–S23). According to our calculations the resulting dianion is the energetically most favored species compared to the three- and fourfold adduct (SI, Table S13).

The anionic systems, $4^{\cdot-}$ and $4^{2-}$, resemble typical porphyrins also regarding their photophysical properties. Porphyrins are characterized by their unique absorption and emission behavior. Especially the prominent Soret (B) bands (blue region) and the Q absorption bands (red region) are typical for this class of compounds. Comparable bands are missing in the spectrum of the non-globally aromatic compound 4, but they appear after reduction for both the radical anion $4^{\cdot-}$ and the dianion $4^{2-}$ (Fig. 6b). This emphasizes the transition of a porphyrinogen to a porphyrin-like character. Conventional porphyrins typically show high absorbances for their Soret bands, whereas their Q bands are in comparison quite weak[73–76]. This turns out differently for K[4] and particularly for K₂[4], which absorb very efficiently at long wavelengths. The maximum molar absorption coefficient of the single absorption band of K₂[4] in the Q band region at 855 nm reaches $2.1 \cdot 10^5$ L mol⁻¹ cm⁻¹. In comparison, undoped representatives of conventional porphyrinoids like tetraphenylporphyrin (TTP) are characterized by a series of Q bands with the most pronounced one being at $1.87 \cdot 10^4$ L mol⁻¹ cm⁻¹ [77]. The absorption of its Soret band on the other hand with its maximum at 406 nm is somewhat lower in relation ($1.1 \cdot 10^5$ L mol⁻¹ cm⁻¹), compared to the Soret band of the benchmark TTP at $4.6 \cdot 10^5$ L mol⁻¹ cm⁻¹ [77,78]. In the radical species K[4] the situation is reversed, so the relative peak heights correspond to those of typical porphyrins with a higher intensity for the Soret band at 365 nm, with $\varepsilon = 8.1 \cdot 10^4$ L mol⁻¹ cm⁻¹, and a still pronounced Q band at 1210 nm, with $\varepsilon = 5.7 \cdot 10^4$ L mol⁻¹ cm⁻¹. K₂[4] additionally shows weak fluorescence emission around a maximum at 902 nm ($\Phi_f = 0.01$) in the near-infrared (NIR) region, with a small Stokes shift of 77.3 meV (Fig. 6c).

The absorption behavior of both anions can be interpreted by means of TD-DFT calculations. The main absorption band of $4^{2-}$ at lower energies (Q band) can be assigned to two nearly degenerated $S_0 \to S_1$ (1.487 eV) and $S_0 \to S_2$ (1.499 eV) excitations. They occur from the HOMO to a degenerate orbital pair composed of LUMO and LUMO+1. This behavior is also described by the NTO picture. Those two transitions have oscillator strengths of around 0.6 and can be assigned to the experimentally observed Q bands (SI, Table S10). We see a similar picture for the band in the blue region where we have again two transitions, namely $S_0 \to S_{14}$ and $S_0 \to S_{15}$ with energies of 3.552 and 3.563 eV and oscillator strengths of 0.671 and 0.690, respectively. These represent the Soret absorption band (Fig. 6f).

The intensity of the Q bands compared to conventional pyrrole-based-porphyrins can be explained by examination of the involved orbitals and symmetry of the respective macrocycles. For porphyrins we see that the Q and Soret bands are described by excitations of a mixture of the four orbitals HOMO−1, HOMO, LUMO and LUMO+1 (SI, Table S11). Because of the fact that excitations from HOMO to LUMO or HOMO−1 to LUMO+1 are nearly equal in energy ($D_{2h}$ point group), we can make two degenerate linear combinations to generate the minus (−) and plus (+) states which are of $B_{3u}$ symmetry. The transition dipole moment vectors (y coordinate for $D_{2h}$) are parallel and have similar dimensions, which leads to a pseudoparity forbidden antiparallel $B_{3u}^-$ (Q band) transition and a parallel $B_{3u}^+$ (Soret band) transition. The same can be applied for the $B_{2u}$ transitions (HOMO → LUMO+1 and HOMO−1 → LUMO). This leads to significantly lower oscillator strengths of the Q bands[73]. As described, our bowl-shaped dianionic species $4^{2-}$ ($C_{4v}$ symmetry) is distorted from the classical porphyrin structure. This leads to a large energetical splitting of the HOMO and HOMO−1 orbitals (Figure S84). This can be ascribed to the fact that the boron atoms, which have a higher atom energy[79] than carbon atoms in TPP, contribute to the HOMO while the HOMO−1 has a nodal plane at each boron center. This leads to a larger elevation of the HOMO in $4^{2-}$ in comparison to TPP. Thus, mixing of these states is less likely, and the lowest-energy transitions that give rise to the Q bands ($S_1$ and $S_2$) consist of single-orbital contributions. Similar conclusions can be drawn for the monoanionic species $4^{\cdot-}$. Here we also have two nearly degenerated $S_1/S_2$ states that look comparable to those of $4^{2-}$ in the

NTO and MO picture (Fig. 6f and S1, Table S9). We computed excitation energies of 1.018 eV ($D_1$) and 1.048 eV ($D_2$). Compared to the dianion we can conclude that the SOMO−LUMO gap is smaller for $4^{\cdot-}$ than the HOMO−LUMO gap is for $4^{2-}$. Finally, the pair of peaks we detected in the blue spectral region can be ascribed to excitations to the $D_{18}$ and $D_{19}$ states, which show degeneracy with computed energies of 3.566 and 3.569 eV, respectively.

We herein presented tetrathiaporphyrinogen 4 with electron-deficient boron centers in all four *meso*-positions. Despite the fact that this macrocycle exhibits only local aromaticity confined in the individual thiophene rings, it comprises a fully π-conjugated macrocyclic backbone. This is evidenced by its spectroscopic features and supported by DFT calculations. Furthermore, different from the previously known *B*-amino substituted tetraboratetrathiaporphyrinogens IIa,b, the thiophene rings in 4 approach a nearly coplanar conformation. The global antiaromaticity of the macrocycle is thus effectively concealed through the local aromaticity in the constituent thiophene units. This makes it very robust. The porphyrinogen 4 readily binds fluoride anions, which is signaled by changes in the absorption characteristics as well as fluorescence quenching. Facile consecutive one-electron reductions of 4 produce the radical anion $4^{\cdot-}$ and the dianion $4^{2-}$, both of which are globally aromatic. The 18-π-electron aromatic porphyrinoid $4^{2-}$ shows the typical absorption features of conventional porphyrins, however, its low-energy Q bands are significantly more intense in comparison. In addition, $4^{2-}$ displays weak fluorescence emission in the NIR spectral region.

## Methods

### General

All reactions were performed under argon atmosphere using standard Schlenk techniques or an MBraun glovebox.

### NMR spectroscopy

NMR spectra were recorded at 298 K on a Bruker Avance III (operating at ¹H: 300 MHz, ¹¹B: 96 MHz) or a Bruker Avance 500 FT NMR spectrometer (operating at ¹H: 500 MHz, ¹¹B: 160 MHz, ¹³C: 126 MHz) at 296 K. Chemical shifts (δ) were referenced to residual protic impurities in the solvent (¹H) or the deuterated solvent itself (¹³C) and reported relative to external SiMe₄ (¹H, ¹³C) or BF₃·OEt₂ (¹¹B) standards, respectively.

### High resolution mass spectrometry

Mass spectra were obtained with the use of a Thermo Scientific Exactive Plus Orbitrap MS system with electron spray ionization (ESI) or by liquid injection field desorption ionization (LIFDI).

### Elemental analysis

Elemental analyses were performed on an Elementar vario MICRO cube elemental analyzer.

### Photophysics

The optical measurements were performed in standard quartz cuvettes (1 cm × 1 cm cross-section). UV-vis absorption spectra were recorded using a Perkin Elmer LAMBDA 465 UV-vis spectrophotometer or a Mettler Toledo UV7 spectrophotometer in a nitrogen-atmosphere in an MBraun glovebox. The measurements reaching into the NIR-region of the spectrum were recorded using a Perkin Elmer LAMBDA 1050 + UV/Vis/NIR spectrophotometer. The emission spectra were recorded using an Edinburgh Instruments FLS920 spectrometer equipped with a double monochromator for both excitation and emission, operating in right-angle geometry mode, and all spectra were fully corrected for the spectral response of the instrument. Fluorescence quantum yields were measured using a calibrated integrating sphere from Edinburgh Instruments combined with the FLS920 spectrometer described above.

## Voltammetry

All cyclic voltammetry (CV) and differential pulse voltammetry (DPV, square wave) experiments were conducted in a nitrogen-filled glovebox using a Gamry Interface 1010B potentiostat. A standard three-electrode cell configuration was employed using a platinum disk working electrode ($d = 3$ mm), a platinum wire counter electrode, and a silver wire reference electrode separated by a Vycor® frit, serving as a pseudo-reference electrode. The redox potentials are referenced to the ferrocene/ferrocenium ([Fc/Fc$^+$]) redox couple as an internal standard. Tetra-$n$-butylammonium hexafluorophosphate ([$n$-Bu$_4$N] [PF$_6$]) was employed as the supporting electrolyte with a concentration of 0.1 mol/L.

## X-ray Crystallography

Crystals suitable for single-crystal X-ray diffraction were selected and coated in perfluoropolyether oil. Diffraction data were collected on Bruker X8 Apex II 4-axis-κ-goniometer diffractometer with APEX II CCD area detector using Mo-K$_\alpha$ radiation or Bruker AXS D8-Quest 4-axis-κ-goniometer diffractometer with Photon II CMOS area detector using Mo-K$_\alpha$ radiation. The crystals were cooled using Oxford Cryostreams low-temperature devices. Data were collected at 100 K.

## EPR Spectroscopy

EPR measurements at X-band (9.37 GHz for **1**$^{\bullet-}$, **4**$^{\bullet-}$ / 9.85 GHz for **2**$^{\bullet-}$, **3**$^{\bullet-}$; microwave power: 2 mW for **1**$^{\bullet-}$, **4**$^{\bullet-}$ / 0.2 mW for **2**$^{\bullet-}$, **3**$^{\bullet-}$) were carried out using a Bruker ELEXSYS E580 CW EPR spectrometer. Modulation amplitude: 0.5 G; Conversion time: 60 ms; modulation frequency: 100 kHz. The spectral simulations were performed using MATLAB 9.8.0.1323502 (R2020a) or 9.13.0.2105380 (R2022b) and the EasySpin 5.2.35 toolbox.

## Synthesis of 4

**Route I via [2 + 2] macrocyclization.** A solution of LDA was prepared by addition of $n$-BuLi (1.20 mmol, 750 μL) in hexane to a solution of di-$iso$-propylamine (121 mg, 1.20 mmol) in THF (1 mL) at −30 °C. The mixture was stirred for 30 min. This solution was added to a mixture of **2** (328 mg, 480 μmol) in THF (5 mL) at −78 °C. The solution was stirred for 4 h maintaining the temperature. The green suspension was diluted with THF (20 mL). A cold (−78 °C) solution of **10** (275 mg, 480 μmol) in THF (25 mL) was added dropwise over the course of 2 h at −78 °C. It was stirred for 3 d, while the cooling bath was allowed to warm up to room temperature. Afterwards, the solvent was removed *in vacuo* and the residue was redissolved in DCM (10 mL) and chlorotrimethylsilane (120 mg, 1.10 mmol) was added. It was stirred for 1 h. Then, the mixture was worked up aqueously and the product was extracted with DCM, washed with brine and water, dried over Na$_2$SO$_4$, and filtered off. The solvent was removed under reduced pressure. The crude product was purified by column chromatography (silica, 3% DCM in PE). Subsequently, the solid product was recrystallized out of ethylacetate to yield yellow needles. Yield: 347 mg (293 μmol, 61%).

**Route II via [3 + 1] macrocyclization.** A solution of LDA was prepared by addition of $n$-BuLi (1.20 mmol, 750 μL) in hexane to a solution of di-$iso$-propylamine (121 mg, 1.20 mmol) in THF (1 mL) at −30 °C. The mixture was stirred for 30 min. This solution was added to a mixture of **1** (190 mg, 500 μmol) in THF (5 mL) at −78 °C. The solution was stirred for 4 h maintaining the temperature. The purple solution was diluted with THF (20 mL). A cold (−78 °C) solution of **7** (434 mg, 500 μmol) in THF (25 mL) was added dropwise over the course of 1 h at −78 °C. It was stirred for 3 d, while the cooling bath was allowed to warm up to room temperature. Afterwards, the solvent was removed *in vacuo* and the residue was redissolved in DCM (10 mL) and chlorotrimethylsilane (163 mg, 1.50 mmol) was added. It was stirred for 1 h. Then, the mixture was worked up aqueously and the product was extracted with DCM, washed with brine and water, dried over Na$_2$SO$_4$, and filtered off. The

solvent was removed under reduced pressure. The crude product was purified by column chromatography (silica, 3% DCM in PE). Subsequently, the solid product was recrystallized out of ethylacetate to yield yellow needles. Yield: 157 mg (132 μmol, 27%). For complete methods please refer to the Supplementary Information.

## Data availability

All data are available from the corresponding author upon request. For experimental details, procedures and spectra see supplementary files. Source data of the spectra are available at the Figshare repository (https://doi.org/10.6084/m9.figshare.28210820)[80]. Crystallographic data have been deposited with the Cambridge Crystallographic Data Center as supplementary publication no. CCDC-2333888 (**4**), CCDC-2333889 (**9**), CCDC-2333890 (**2**), CCDC-2333891 (**K$_2$[4]**), CCDC-2333892 (**K[4]**) and CCDC-2333893 (**10**). These data can be obtained free of charge from the Cambridge Crystallographic Data Center via www.ccdc.cam.ac.uk/data_request/cif. The cartesian coordinates for the calculated structures used in this study are available at the ioChem-BD Computational Chemistry repository (https://doi.org/10.19061/iochem-bd-6-436)[81]. Source data are provided with this paper.

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

## Acknowledgements

Financial support by the Deutsche Forschungsgemeinschaft (DFG, German Research Foundation) through the Heisenberg Grant HE 6171/9-1, 468457264, and 466754611 is gratefully acknowledged.

## Author contributions

M.B. performed the syntheses, cyclic voltammetric measurements and photophysical measurements. J.K. performed the computational studies. M.B. and J.S.S. performed X-ray crystal structure analyses. A.L. and N.A.R. performed syntheses. I.K. performed the EPR measurements. H.B. supervised the EPR experiments. B.E. supervised the computational studies. H.H. provided the resources, conceptualized the research, performed the funding acquisition, supervision, and administration of the project. M.B. and H.H. wrote the original paper draft, and the manuscript was reviewed and edited by all co-authors.

## Funding

## Competing interests

The authors declare no competing interest.
