## [Transparent Peer Review file · Nature Communications]

A Fully Conjugated *Meso*-Boron-Substituted Porphyrinoid Combining Lewis Acidity with Redox-Switchable Aromaticity

Corresponding Author: Professor Holger Helten

Version 0:

Reviewer comments:

Reviewer #1

(Remarks to the Author)

The authors here reported the synthesis and comprehensive characterization of a fully conjugated porphyrinoid with electron-deficient boron-bridged in all four meso-positions (tetraboratetraphiaporphyrinogen 4). The radical anion (4^{•-}) and the dianion (4²⁻) were also prepared via chemical reductions and characterized by single-crystal X-ray diffraction, revealing a global aromaticity switching between neutral and reduced states. Overall, this manuscript presents an interesting work on boron-doped thiophene-based porphyrinoid and is thus recommended for publication after revision.

1. The author discussed the spin densities of four anions (1^{•-}, 2^{•-}, 3^{•-}, 4^{•-}) and mentioned that 'these observations are consistent with an increased delocalization of the spin density with increasing size of the system', the spin distributions in Supplementary Information.
2. How is the stability of radical anion (4^{•-}) and the dianion (4²⁻), and whether the global aromaticity is conducive to the stabilization of these reduced species?
3. An interesting phenomenon mentioned in the article is that, compared to undoped porphyrinoids (tetraphenylporphyrin, TTP), the dianion (4²⁻) exhibits significantly stronger absorption in the Q bands. Why?
4. In the abstract the authors mentioned that "the bowl-shaped dianion constitutes the first example of a thiophene-based porphyrinoid with a metal cation in its coordination sphere." A more detailed analysis of the structural characteristics is needed in the main text to support this claim. For example, bowl depth variation from 4^{•-} to 4²⁻, bond length between sulfur and potassium, and even molecular stacking structures.
5. Why did 4 only bind two fluoride ions?

Reviewer #2

(Remarks to the Author)

This paper describes a porphyrinoid (4) which uniquely incorporates boron atoms into its molecular framework. The porphyrinogen, as synthesized, is antiaromatic with local aromaticity at the individual thiophene units which make up its backbone. Upon 1 and 2 e⁻ reduction, this molecule experiences global aromaticity and displays porphyrin-like character. The paper covers the complete synthesis and characterization of this molecule and its negatively charged counterparts. This includes electrochemical studies, X-ray diffraction analysis, EPR, photophysical studies, TD-DFT calculations, and UV-vis data showing the macrocycle's ability to bind fluoride ions. The authors conclude by saying this type of molecule, given its observed photophysical characteristics, may be further tuned to create more efficient NIR emitters. This paper does an excellent job of laying out all necessary data to characterize the redox-dependent aromatic nature of this novel, boron containing porphyrinoid. The paper is very readable, thorough, and provides significant extratextual support via the Supporting Information document. This contribution describes an interesting advance in porphyrinoid chemistry. Publication in Nature Communications is recommended.

Minor comments:

- 1) "...while its low-energy Q-bands are significantly stronger." This language is imprecise.
- 2) What is meant by "weak NIR emission"? State the emission quantum yield that is reported later in the manuscript.
- 3) Figure 2 is sufficient. The descriptive synthesis should be placed in the Supporting Information.
- 4) The final statement of the conclusion ("...optimization of the emissive strengths could render this class of compounds promising candidates for future applications as efficient NIR emitters. Such materials with high quantum yields are still

scarce”) appears unnecessary as it is unrelated to the data presented in the paper.

Version 1:

Reviewer comments:

Reviewer #1

(Remarks to the Author)

The new version of this manuscript is significantly improved and well-resolved my concerns and I thus recommend for publication.

Point-by-Point Response to the Reviewers' Comments

Reviewer #1 (Remarks to the Author):

The authors here reported the synthesis and comprehensive characterization of a fully conjugated porphyrinoid with electron-deficient boron-bridged in all four meso-positions (tetraboratetrathiaporphyrinogen 4). The radical anion ($4\bullet^-$) and the dianion (4^{2-}) were also prepared via chemical reductions and characterized by single-crystal X-ray diffraction, revealing a global aromaticity switching between neutral and reduced states. Overall, this manuscript presents an interesting work on boron-doped thiophene-based porphyrinoid and is thus recommended for publication after revision.

1. The author discussed the spin densities of four anions ($1\bullet^-$, $2\bullet^-$, $3\bullet^-$, $4\bullet^-$) and mentioned that 'these observations are consistent with an increased delocalization of the spin density with increasing size of the system', the spin distributions in Supplementary Information.

RESPONSE: We thank the reviewer for this remark. It led us to inspect the spin densities more precisely. We have now added them to the SI (Fig. S87). The spin density becomes progressively more localized at the boron centers with increasing number of borane moieties. In the macrocycle it is uniformly distributed over all four boron atoms (and to a small extent at the sulfur atoms). We have now briefly discussed this in the manuscript above Fig. 5.

2. How is the stability of radical anion ($4\bullet^-$) and the dianion (4^{2-}), and whether the global aromaticity is conducive to the stabilization of these reduced species?

RESPONSE: The dianion is highly sensitive towards air and moisture. We have now added a note on this to the manuscript (below Fig. 3, highlighted in yellow). This, however, prevents us from being able to provide experimental evidence of the influence of the global aromaticity on stability. The radical anion is relatively robust in comparison, but also undergoes oxidation over time. The neutral macrocycle on the other hand is fully bench stable.

3. An interesting phenomenon mentioned in the article is that, compared to undoped porphyrinoids (tetraphenylporphyrin, TTP), the dianion (4^{2-}) exhibits significantly stronger absorption in the Q bands. Why?

RESPONSE: For common porphyrins the intensities of the Q bands are so low because of dipole cancellation due to its highly symmetric shape (D_{2h}) and correlation of the involved orbitals. Our macrocycles possess lower symmetry (C_{4v}) because of their bowl-type structure, and the lowest-energy transitions that give rise to the Q bands (S_1 and S_2) consist of transitions with single-orbital contributions of HOMO \rightarrow LUMO and HOMO \rightarrow LUMO+1 without mixing of different contributing orbitals because of an enlarged energy gap between HOMO and HOMO-1 in comparison to TPP. This results in higher oscillator strengths and thus more intense Q bands. We have now included a more detailed discussion in the main text (highlighted passage before the Conclusion).

4. In the abstract the authors mentioned that "the bowl-shaped dianion constitutes the first example of a thiophene-based porphyrinoid with a metal cation in its coordination sphere." A more detailed analysis of the structural characteristics is needed in the main text to support this claim. For example, bowl depth variation from 4^- to 4^{2-} , bond length between sulfur and potassium, and even molecular stacking structures.

RESPONSE: We have now added a more detailed discussion to the main text (highlighted passage to the left of Fig. 5). Especially the bond angle towards the potassium cation shows

the typical picture of thiophene-metal bonding. Alongside bond-lengthening of the inner-thiophene S-C Bond, indicating electron-density transfer from sulfur to the cation, the S-K bond is shorter than the sum of the van-der-Waals radii of both atoms. Additionally, the bowl-depth increases going from the radical to the dianionic species. No notable stacking effects or intermolecular interactions between the macrocycles were observed.

5. Why did 4 only bind two fluoride ions?

RESPONSE: We have now performed DFT calculations of adducts with 1-4 fluoride ions (SI, Table S13). These reveal that the species with 2 fluoride anions bound in an anti-fashion is the most energetically favorable. This suggests that the addition of more than 2 fluoride ions is prevented by Coulombic repulsion. We have now mentioned our new computations briefly in the main text (right column, below Fig. 6).

Reviewer #2 (Remarks to the Author):

This paper describes a porphyrinoid (4) which uniquely incorporates boron atoms into its molecular framework. The porphyrinogen, as synthesized, is antiaromatic with local aromaticity at the individual thiophene units which make up its backbone. Upon 1 and 2 e-reduction, this molecule experiences global aromaticity and displays porphyrin-like character. The paper covers the complete synthesis and characterization of this molecule and its negatively charged counterparts. This includes electrochemical studies, X-ray diffraction analysis, EPR, photophysical studies, TD-DFT calculations, and UV-vis data showing the macrocycle's ability to bind fluoride ions. The authors conclude by saying this type of molecule, given its observed photophysical characteristics, may be further tuned to create more efficient NIR emitters.

This paper does an excellent job of laying out all necessary data to characterize the redox-dependent aromatic nature of this novel, boron containing porphyrinoid. The paper is very readable, thorough, and provides significant extratextual support via the Supporting Information document. This contribution describes an interesting advance in porphyrinoid chemistry. Publication in Nature Communications is recommended.

Minor comments:

1) "...while its low-energy Q-bands are significantly stronger." This language is imprecise.

RESPONSE: We have now formulated the statement more precisely.

2) What is meant by "weak NIR emission"? State the emission quantum yield that is reported later in the manuscript.

RESPONSE: We have complemented the quantum yield, as suggested.

3) Figure 2 is sufficient. The descriptive synthesis should be placed in the Supporting Information.

RESPONSE: The synthesis descriptions of all macrocycles were merged and significantly shortened in the main text, as suggested (see first paragraph of the Results part, highlighted in yellow).

4) The final statement of the conclusion ("...optimization of the emissive strengths could render this class of compounds promising candidates for future applications as efficient NIR emitters. Such materials with high quantum yields are still scarce") appears unnecessary as it is unrelated to the data presented in the paper.

RESPONSE: We have removed this statement from the conclusion, as suggested.

Point-by-Point Response to the Reviewers' Comments

Reviewer #1 (Remarks to the Author):

The new version of this manuscript is significantly improved and well-resolved my concerns and I thus recommend for publication.

RESPONSE: We thank the reviewer for their positive assessment of our work.